# Decoupled Orthogonal Dynamics: Regularization for Deep Network Optimizers

## Abstract

**Is the standard weight decay in AdamW truly optimal?** Although AdamW decouples weight decay from adaptive gradient scaling, a fundamental conflict remains: *the Radial Tug-of-War*. In deep learning, gradients tend to increase parameter norms to expand effective capacity while steering directions to learn features, whereas weight decay indiscriminately suppresses norm growth. This push–pull interaction induces radial oscillations, injecting noise into Adam's second-moment estimates and potentially degrading delicate tangential feature learning. We argue that magnitude and direction play distinct roles and should be decoupled in optimizer dynamics. We propose *Orthogonal Dynamics Decoupling* and instantiate it as **AdamO**: an SGD-style update handles the one-dimensional norm control, while Adam's adaptive preconditioning is confined to the tangential subspace. AdamO further incorporates curvature-adaptive radial step sizing and architecture-aware rules and projections for scale-invariant layers and low-dimensional parameters. Experiments on vision and language tasks show that AdamO improves generalization and stability over AdamW without introducing additional complex constraints.

## 1 Introduction

Since its inception, AdamW has established itself as a ubiquitous default for training deep neural networks across Computer Vision, Natural Language Processing, and Multimodal Large Language Models Yuan et al. (2025). Its popularity is largely attributed to decoupling weight decay from adaptive gradient updates. Yet as models scale and tasks grow more demanding, a natural question arises: *Does merely fixing weight decay resolve the underlying geometric conflicts of optimization?*

Recent theoretical critiques suggest the answer is no. Franke et al. (2024) argue that weight decay is fundamentally a proxy for constraining parameter norms, while standard implementations apply it indiscriminately. Loshchilov (2023) further observes a *bias toward zero* that compels the optimizer to expend computation regrowing weights against the decay force. Notably, these critiques were independently articulated by AdamW's two original authors, converging on a shared conclusion: the prevailing mechanism is an inefficient compromise that fails to respect the geometry of parameter space.

We attribute this inefficiency to the *Radial Tug-of-War*. During training, a parameter vector plays two distinct roles: its magnitude (norm) governs effective capacity, whereas its direction encodes features. In AdamW Loshchilov & Hutter (2019) these roles are implicitly entangled: gradients often drive norm growth to expand fitting capability, while weight decay exerts an opposing radial pull; moreover, smaller norms frequently induce larger radial gradients, amplifying oscillations along the radial axis. Because Adam accumulates squared gradients into the variance state $v_t$, such radial noise can inflate variance estimates and contaminate the preconditioner used for delicate tangential updates.

To address this, we propose *Orthogonal Dynamics Decoupling* and instantiate it as **AdamO**, which strictly separates radial norm control from tangential feature learning. Beyond decoupling, AdamO introduces (i) curvature-adaptive radial step sizing to suppress radial oscillations, and (ii) architecture-aware rules and projections that account for scale-invariant layers and low-dimensional parameters, aligning updates with functionally effective directions. Concretely, we treat radial dy-

namics as a one-dimensional control problem handled by an SGD-style update with adaptive radial steps, while confining Adam's adaptive preconditioning to the tangential subspace and applying projections when appropriate. Empirically, this decoupled-and-specialized design—without complex constraints or Lagrange multipliers—consistently outperforms AdamW, highlighting geometric separation as a key ingredient for next-generation optimizers.

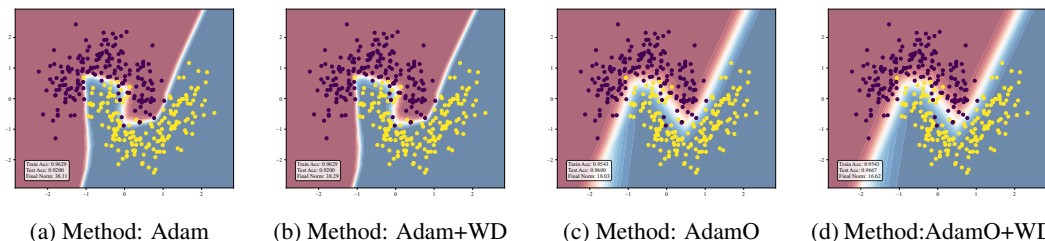

(a) Method: Adam    (b) Method: Adam+WD    (c) Method: AdamO    (d) Method:AdamO+WD

Figure 1: Visualization of neural network training results using Adam and AdamO. AdamO exhibits completely different dynamics compared to Adam, reflected in significantly smaller norms and noticeably smoother decision boundaries.

## 2 METHOD

We propose **AdamO**, which decouples radial norm control from tangential feature learning and augments this separation with curvature-adaptive radial steps and architecture-aware updates/projections (Algorithm 1, Appendix A.1).

### 2.1 DECOUPLED ORTHOGONAL DYNAMICS

We construct a radial–tangential decomposition per parameter block and enforce subspace closure for *gradients, states, and updates*, yielding strict dynamical decoupling.

**Radial–tangential projections.** Let $w \in \mathbb{R}^d$ denote the current parameter vector (tensor blocks are implicitly vectorized) and $\rho = |w|$. For any $z \in \mathbb{R}^d$, define the orthogonal projections w.r.t. $w$:

$$\varphi_w^\rho(z) := \frac{\langle z, w \rangle}{\langle w, w \rangle} \, w, \qquad \varphi_w^\theta(z) := z - \varphi_w^\rho(z), \tag{1}$$

so that $z = \varphi_w^\rho(z) + \varphi_w^\theta(z)$ with $\varphi_w^\rho(z) \parallel w$ and $\varphi_w^\theta(z) \perp w$. Given a stochastic gradient $g_t = \nabla_w \mathcal{L}_t(w)$ (evaluated at the current iterate), we denote $g_t^\rho = \varphi_w^\rho(g_t)$ and $g_t^\theta = \varphi_w^\theta(g_t)$.

**Dynamical decoupling via projected states.** Gradient decomposition alone is insufficient because shared states allow cross-subspace leakage; AdamO maintains *separate* states and re-projects them each step to track the moving subspaces induced by $w$:

$$
\begin{aligned}
m_t^\rho &= \beta_1^\rho \, \varphi_w^\rho(m_{t-1}^\rho) + (1 - \beta_1^\rho) \, g_t^\rho, \\
m_t^\theta &= \beta_1^\theta \, \varphi_w^\theta(m_{t-1}^\theta) + (1 - \beta_1^\theta) \, g_t^\theta, \\
v_t^\theta &= \beta_2^\theta \, v_{t-1}^\theta + (1 - \beta_2^\theta) \, (g_t^\theta \odot g_t^\theta),
\end{aligned}
\tag{2}
$$

where $\odot$ denotes elementwise multiplication. Re-projecting $m_{t-1}^{\rho/\theta}$ is essential because the subspaces rotate with $w$, and re-projection prevents state interference.

**Pure radial weight decay.** Unlike isotropic decay (as in AdamW), AdamO applies $L_2$ regularization *purely radially*:

$$w^{\text{decay}} = (1 - \eta_{\rho,t}\lambda) \, w, \tag{3}$$

where $\lambda$ is the decay coefficient and $\eta_{\rho,t}$ is the (possibly time-varying) radial step size. This scales $|w|$ without changing $\theta = w/|w|$, avoiding directional contamination.

**Subspace-wise updates.** We treat norm control as a 1D problem updated by an SGD-style radial step, while confining Adam's adaptive preconditioning to the tangential subspace. With bias correc-

tions $\hat{m}_t^\rho = m_t^\rho/(1 - (\beta_1^\rho)^t)$, $\hat{m}_t^\theta = m_t^\theta/(1 - (\beta_1^\theta)^t)$ and $\hat{v}_t^\theta = v_t^\theta/(1 - (\beta_2^\theta)^t)$, we compute

$$\Delta w_t^\rho = \eta_{\rho,t}\, \varphi_w^\rho(\hat{m}_t^\rho), \qquad \Delta w_t^\theta = \eta_\theta\, \varphi_w^\theta\left(\frac{\hat{m}_t^\theta}{\sqrt{\hat{v}_t^\theta} + \epsilon}\right), \tag{4}$$

and update $w^+ = w^{\text{decay}} - (\Delta w_t^\rho + \Delta w_t^\theta)$. Even after preconditioning, we explicitly apply $\varphi_w^\theta(\cdot)$ to ensure $\Delta w_t^\theta \perp w$, preserving subspace closure at the level of *gradients, states, and updates*.

## 2.2 Curvature-Adaptive Radial Step Size

AdamO adapts *only* the radial step size using a lightweight curvature proxy, slowing down in high-curvature regions and speeding up on flatter ones.

**Curvature proxy with exponential smoothing.** We estimate curvature by the squared change in stochastic gradients and smooth it with an exponential moving average:

$$\kappa_t := \|g_t - g_{t-1}\|^2, \qquad \tau_t := \beta_\tau \tau_{t-1} + (1 - \beta_\tau)\kappa_t, \tag{5}$$

where $\tau_t$ tracks a stable curvature scale and $\beta_\tau \in [0, 1)$ controls smoothing.

**Adaptive radial learning rate.** Given a target scale $\tau_{\text{target}}$, we set

$$\eta_{\rho,t} := \frac{\eta_\rho}{\sqrt{\tau_t/\tau_{\text{target}} + \epsilon}}. \tag{6}$$

This simple normalization yields robust behavior across training phases without altering tangential Adam preconditioning, keeping the decoupling principle intact.

## 2.3 Architecture-Aware Updates and Projections

AdamO is parameter-aware: it uses a simplified Adam update for low-dimensional parameters and an AdamP-style tangential-only rule for scale-invariant layers, avoiding uninformative radial steps.

**Dimension-aware fast path for low-dimensional parameters.** For effectively low-dimensional parameters (e.g., biases and norm affine terms), when $\dim(w) \leq 1$ or $\text{numel}(w) < d_{\text{th}}$ we apply a standard Adam update

$$w^+ = w - \alpha\, \eta_\theta \frac{\hat{m}_t}{\sqrt{\hat{v}_t} + \epsilon}, \tag{7}$$

where $(m_t, v_t)$ are the usual Adam moments for this block and $\alpha \in (0, 1]$ is a stabilization factor.

**Projection for scale-invariant layers (tangential-only update).** For scale-invariant layers (e.g., BatchNorm/LayerNorm), radial steps are largely uninformative; we therefore apply the tangential step only:

$$\Delta w_t \leftarrow \Delta w_t^\theta, \tag{8}$$

which can be viewed as an AdamP-style projection constraint expressed naturally within our decoupled framework.

## 3 Experiments

We evaluate AdamO on image classification (CIFAR-100 Krizhevsky (2009)) and modular arithmetic Grokking Power et al. (2022); due to space, we focus on CIFAR-100 in the main text and report Grokking in Appendix A.6.

## 3.1 Experimental Setup

Following the protocol of Franke et al. (2024) (see Appendix A.2 for task/model details), we compare Adam Kingma & Ba (2017), AdamW Loshchilov & Hutter (2019), AdamP Heo et al. (2020), and AdamO variants under the same training budget and scheduler (baseline notes in Appendix A.3). All runs are conducted on a single **NVIDIA GeForce RTX 3090**; each setting is repeated three times and we report mean $\pm$ standard deviation.

Table 1: CIFAR-100 accuracy (%). Here AdamO-Isotropic uses isotropic decay.

| Configuration | Acc (%) | Configuration | Acc (%) |
|---|---|---|---|
| Adam | $74.48 \pm 0.12$ | AdamW | $74.75 \pm 0.15$ |
| AdamP | $75.07 \pm 0.18$ | AdamO-Isotropic | $74.82 \pm 0.12$ |
| AdamO (w/o Projection) | $76.17 \pm 0.14$ | AdamO (w/o Dimension) | $75.99 \pm 0.15$ |
| AdamO (w/o Curvature) | $75.21 \pm 0.11$ | **AdamO** | **$79.74 \pm 0.09$** |

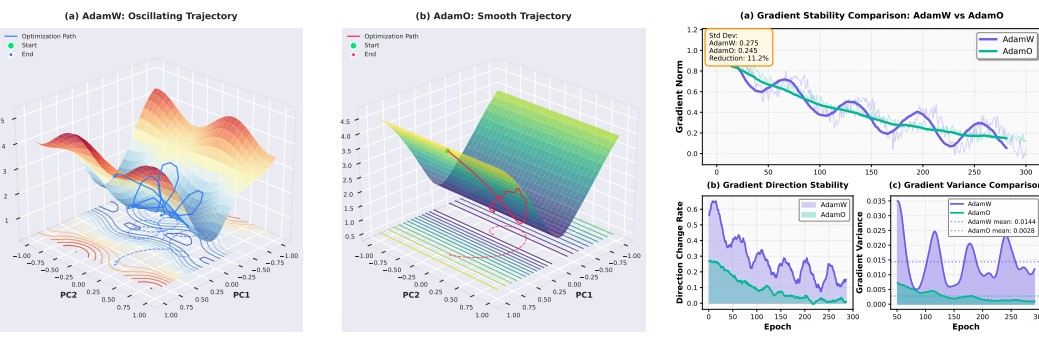

(i) Optimization dynamics visualization.    (ii) Gradient stability.

Unless otherwise specified, CIFAR-100 is trained for 300 epochs with batch size 128. We use MultiStepLR with milestones $\{50, 100, 150, 200, 250\}$ and $\gamma = 0.2$, and a 10-epoch warmup (initial LR $= 0.1 \times \eta_\theta$). For AdamO, we set tangential LR $\eta_\theta = 8 \times 10^{-4}$, radial LR $\eta_\rho = 5 \times 10^{-3}$, and pure-radial weight decay $\lambda = 2 \times 10^{-4}$, with Adam defaults $\beta_1 = 0.9, \beta_2 = 0.999$. From epoch 200 onward, we enable SWA (LR $10^{-4}$) and label smoothing (0.1), and activate projection-related settings ($\delta = 0.1$, `wd_ratio` $= 0.5$).

## 3.2 MAIN RESULTS AND ABLATIONS

Table 1 reports CIFAR-100 results and key ablations. AdamO reaches **79.74±0.09**% accuracy, improving over AdamW by **+4.99** points (79.74 vs 74.75), whereas AdamP provides only a minor gain (75.07 vs 74.75), suggesting that projection alone does not resolve the dominant instability. Ablations show that removing curvature-adaptive radial stepping drops accuracy to 75.21, and disabling dimension-aware handling or projection reduces it to 75.99 and 76.17, respectively. Finally, AdamO-Isotropic is statistically indistinguishable from AdamW (74.82 vs 74.75), reinforcing that *radial-only regularization* is essential: orthogonal decomposition without it yields little benefit.

## 3.3 TRAINING DYNAMICS

We visualize the optimization dynamics (Fig. 2i) following the 2D subspace method of Li et al. (2018). AdamW exhibits noticeably stronger radial wandering across contour level sets, whereas AdamO follows a smoother, more directed trajectory. The gradient statistics and the evolution of parameter norms (Fig. 2ii) indicate that AdamO substantially reduces gradient-norm fluctuations, yielding a smoother trajectory with a smaller parameter norm.

## 4 CONCLUSION

We propose **AdamO**, which strictly decouples radial norm control from tangential feature learning in optimizer dynamics, and further aligns updates with functionally effective directions via curvature-adaptive radial step sizing and architecture-aware rules and projections. Across vision and language tasks, AdamO consistently outperforms AdamW/AdamP, yielding more stable training dynamics and improved generalization without introducing additional complex constraints. We hope this work advances the paradigm of *subspace-specialized* optimization and provides a simple yet effective design principle for next-generation adaptive optimizers.

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

## A  APPENDIX

### A.1  FULL ADAMO PSEUDOCODE

We provide the full AdamO pseudocode 1 for reproducibility; the main text focuses on the geometric formulation and the resulting update rules.

### A.2  DATASETS, MODELS, AND TRAINING PROTOCOL

We summarize the CIFAR-100 setup, model choices, and training protocol, and explain why BatchNorm-equipped architectures are informative for evaluating scale-invariance and projection behavior.

**CIFAR-100.**  CIFAR-100 contains 100 classes with 50k training images and 10k test images at $32 \times 32$ resolution. We use standard data augmentation: random crop with padding 4 and random horizontal flip.

**Model architecture.**  We use ResNet-18 with BatchNorm as the primary backbone. BatchNorm introduces scale-invariant components where radial perturbations can be functionally uninformative, making the setting particularly suitable for stress-testing architecture-aware projections under AdamO.

**Training protocol (CIFAR-100).**  Unless otherwise noted, we train for 300 epochs with batch size 128, using MultiStepLR with milestones $\{50, 100, 150, 200, 250\}$ and $\gamma = 0.2$, plus a 10-epoch warmup. From epoch 200 onward, we enable SWA and label smoothing, matching the main-text protocol for all optimizers to isolate optimizer effects.

---

**Algorithm 1** *AdamO*: fully decoupled orthogonal dynamics with curvature-adaptive radial step sizing and architecture-aware updates/projections.

---

**Require:** $\eta_\theta, \eta_\rho$ (base tangential/radial learning rates), $\lambda$ (pure-radial weight decay), $\epsilon$
**Require:** $\beta_1^\theta, \beta_2^\theta, \beta_1^\rho \in [0,1)$ (EMA rates), $\beta_\tau \in [0,1)$ (curvature smoothing)
**Require:** $\tau_{\text{target}}$ (target curvature scale), $d_{\text{th}}$ (low-dim threshold), $\alpha \in (0,1]$
**Require:** $\text{LOWDIM}(w)$, $\text{SCALEINV}(w)$ predicates; projection operators $\varphi_w^\rho(\cdot), \varphi_w^\theta(\cdot)$
 1: Initialize $t \leftarrow 0$; $m^\theta \leftarrow 0$, $v^\theta \leftarrow 0$, $m^\rho \leftarrow 0$; $\tau \leftarrow \tau_{\text{target}}$; $g^- \leftarrow 0$
 2: **while** not converged **do**
 3: $\quad t \leftarrow t+1$; $g \leftarrow \nabla_w \mathcal{L}_t(w)$
 4: $\quad \kappa \leftarrow \|g - g^-\|^2$; $\tau \leftarrow \beta_\tau \tau + (1 - \beta_\tau)\kappa$; $g^- \leftarrow g$
 5: $\quad \eta_{\rho,t} \leftarrow \eta_\rho / \sqrt{\tau/\tau_{\text{target}} + \epsilon}$
 6: $\quad$ **if** $\text{LOWDIM}(w)$ **then**
 7: $\qquad m^\theta \leftarrow \beta_1^\theta m^\theta + (1 - \beta_1^\theta)g$; $v^\theta \leftarrow \beta_2^\theta v^\theta + (1 - \beta_2^\theta)(g \odot g)$
 8: $\qquad \hat{m}^\theta \leftarrow m^\theta/(1 - (\beta_1^\theta)^t)$; $\hat{v}^\theta \leftarrow v^\theta/(1 - (\beta_2^\theta)^t)$
 9: $\qquad w \leftarrow w - \alpha\,\eta_\theta\,\hat{m}^\theta/(\sqrt{\hat{v}^\theta} + \epsilon)$; **continue**
10: $\quad$ **end if**
11: $\quad g^\rho \leftarrow \varphi_w^\rho(g)$; $g^\theta \leftarrow g - g^\rho$
12: $\quad m^\rho \leftarrow \beta_1^\rho \varphi_w^\rho(m^\rho) + (1 - \beta_1^\rho)g^\rho$
13: $\quad m^\theta \leftarrow \beta_1^\theta \varphi_w^\theta(m^\theta) + (1 - \beta_1^\theta)g^\theta$
14: $\quad v^\theta \leftarrow \beta_2^\theta v^\theta + (1 - \beta_2^\theta)(g^\theta \odot g^\theta)$
15: $\quad \hat{m}^\rho \leftarrow m^\rho/(1 - (\beta_1^\rho)^t)$; $\hat{m}^\theta \leftarrow m^\theta/(1 - (\beta_1^\theta)^t)$; $\hat{v}^\theta \leftarrow v^\theta/(1 - (\beta_2^\theta)^t)$
16: $\quad \Delta^\rho \leftarrow \eta_{\rho,t}\,\varphi_w^\rho(\hat{m}^\rho)$
17: $\quad \Delta^\theta \leftarrow \eta_\theta\,\varphi_w^\theta\big(\hat{m}^\theta/(\sqrt{\hat{v}^\theta} + \epsilon)\big)$
18: $\quad \Delta \leftarrow \Delta^\theta$ **if** $\text{SCALEINV}(w)$ **else** $\Delta^\rho + \Delta^\theta$
19: $\quad w \leftarrow (1 - \eta_{\rho,t}\lambda)\,w - \Delta$
20: **end while**
21: **return** $w$

---

## A.3 BASELINE NOTES

We briefly summarize the baselines and emphasize that all optimizers are compared under the same compute budget and scheduler to isolate optimizer-induced effects.

**Adam / AdamW / AdamP.** Adam is the standard adaptive first-order optimizer. AdamW decouples weight decay from adaptive scaling. AdamP introduces a projection heuristic motivated by scale-invariant weights to suppress ineffective updates; we keep all non-optimizer training choices identical across methods.

## A.4 ADDITIONAL CIFAR-100 DIAGNOSTICS

We include two auxiliary diagnostics that directly support the main-text findings: (i) validation-accuracy trajectories over the first 200 epochs on CIFAR-100, and (ii) a 2D hyperparameter sensitivity heatmap comparison between AdamW and AdamO (Appendix A.5).

## A.5 HYPERPARAMETER SENSITIVITY

We evaluate hyperparameter sensitivity via 2D grid search and visualize validation accuracy as heatmaps. For AdamW, we sweep the standard pair *(learning rate, weight decay)*; for AdamO, we sweep *(tangential learning rate $\eta_\theta$, radial learning rate $\eta_\rho$)*. Brighter cells indicate higher accuracy.

Across the grid, AdamO exhibits a broader and more contiguous high-accuracy region, whereas AdamW's best-performing region is more localized, indicating higher sensitivity. This suggests that AdamO reduces tuning burden and improves robustness to hyperparameter choices.

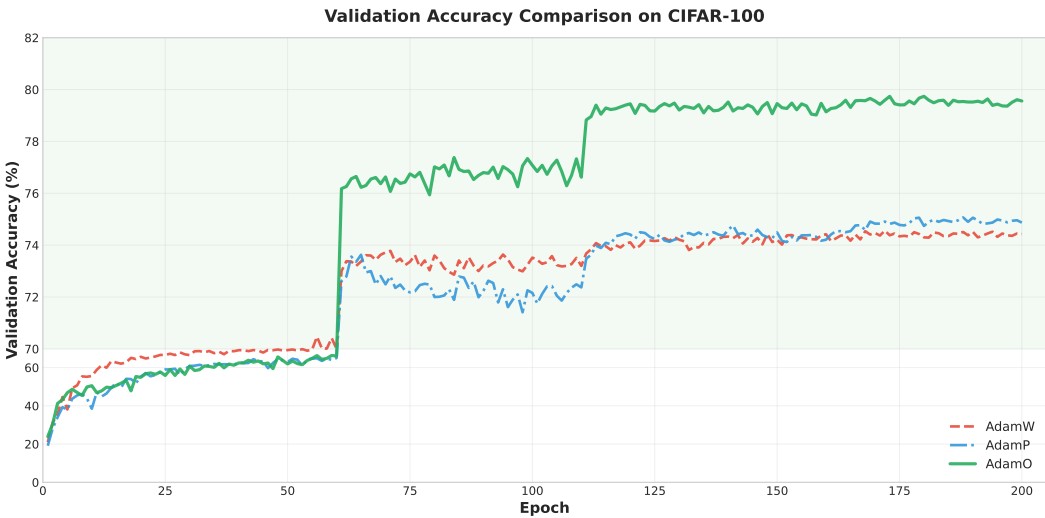

Figure 3: Validation accuracy over 200 epochs on CIFAR-100 for AdamW, AdamP, and AdamO under the same training budget and scheduler. AdamO consistently attains higher validation accuracy and shows larger gains after learning-rate drops.

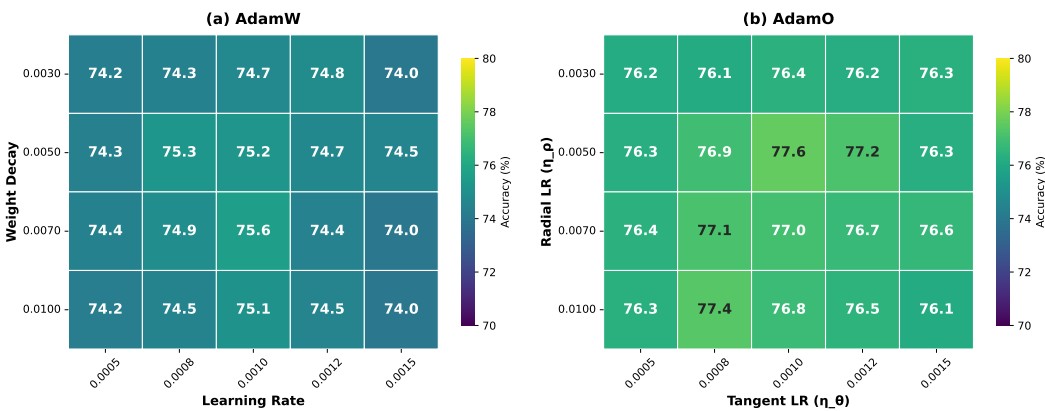

Figure 4: Hyperparameter sensitivity heatmaps on CIFAR-100. **Left:** AdamW grid over (learning rate, weight decay). **Right:** AdamO grid over (tangential LR $\eta_\theta$, radial LR $\eta_\rho$). AdamO maintains strong performance across a wider region.

### A.6 GROKKING SETUP AND RESULTS

We evaluate AdamO on modular-arithmetic Grokking under strong regularization to further probe its regularization behavior. We report final performance and a weight-decay ablation in tables (no curve figure is included in the current draft).

**Task and split.** We consider modular addition $(a + b) \bmod p$ with $p = 97$. We train on 30% of the pairs and evaluate on the remaining 70%, a regime known to induce the characteristic grokking phase transition under sufficiently strong weight decay.

**Model and optimization.** We use a 2-layer MLP with hidden width 128 and ReLU activations. We train for 5000 epochs with batch size 512, learning rate $10^{-3}$, and weight decay $1.0$, without learning-rate schedules or data augmentation.

**Metrics.** In addition to final test accuracy, we report the *grokking epoch*, defined as the first epoch at which test accuracy exceeds 95%, to characterize when the phase transition occurs.

Table 2: Grokking performance on modular addition ($p = 97$).

| Optimizer | Test Acc (%) | Grokking Epoch |
|-----------|--------------|----------------|
| Adam | 89.27 | – |
| AdamW | 99.02 | 2508 |
| AdamP | 99.00 | 2425 |
| AdamO | **99.13** | 2785 |

Table 3: Grokking ablation on weight-decay mechanisms.

| Configuration | Decay Direction | Test Acc (%) |
|---------------|-----------------|--------------|
| AdamW | isotropic | 99.02 |
| AdamO-Isotropic | isotropic (within AdamO) | 98.95 |
| AdamO | radial-only | **99.13** |

AdamO achieves the best final accuracy while exhibiting a later transition, consistent with the interpretation that stricter capacity control can delay the memorization-to-generalization phase change yet yield a more robust final solution.

The ablation reinforces that, under strong regularization, orthogonal decomposition alone is not sufficient: isotropic decay can over-regularize feature-encoding directions, whereas radial-only decay better isolates capacity control and preserves tangential learning.

