# OpenReview forum: "Decoupled Orthogonal Dynamics: Regularization for Deep Network Optimizers"
_ICLR.cc/2026/Workshop/Sci4DL — Sci4DL 2026_

### Official Review · Reviewer_k7kJ · 2026-02-12

**Fit:** 2
**Significance:** 2
**Confidence:** 2

**Summary:**

This paper proposes to decouple the radial and tangential parts (with respect to the parameters) of the momentum in the Adam optimizer, and use two different pre-specified learning rates (SGD for radial + Adam for tangential).

**Strengths:**

- I believe this is a significant contribution to the machine learning field, as it improves a widely-used important optimizer (Adam) elegantly and significantly.
- As the denominator in the tangential dynamics uses the tangential part itself, AdamO is different from (and outperforms) AdamP (which uses the entire weight), increasing the novelty of the method.

**Suggestions:**

- My main concern is the efficiency compared with Adam: effectively, AdamO performs two gradient steps (radial SGD+tangent Adam) in one “total step” (Algorithm 1), while vanilla Adam performs one step. To control the amount of computation in the apple-to-apple comparison, would it be fair to compare AdamO with Adam with doubled steps instead? (Since SGD is lighter than Adam, to mitigate this, a computation throughput/FLOPs comparison would be helpful.)
- As a 2026 paper, this paper does not discuss Muon—would it be promising to to apply Muon to the tangential part of AdamO and observe its performance (making a “MuonO” paper)?
(*Comment:* Even though Muon does not rely on Adam, this paper enables the AdamO dynamics for the embedding matrix (a significant part of small language models), improving the applicability of the paper.)
- BN/LN layers are always 1-dimensional, so I wonder if scale-invariant layers should refer to “layers *preceding* normalization layers”—since the current text does not make this explicit.
- On SGD radial dynamics instead of Adam: I agree that radial subspace is 1-dimensional, but would a quick ablation experiment verification be necessary?

**Overall assessment:**
- For the significance of the contribution, I recommend an acceptance to the workshop.

---

### Official Review · Reviewer_92hm · 2026-02-26

**Fit:** 2
**Significance:** 2
**Confidence:** 3

**Summary:**

This paper proposes a new optimization method. The algorithm is based on Adam, but treats the radial and tangential part of the gradient separately. The authors claim that this results in a better control of the radial component of the parameters during optimization, and they consider additional -decoupled- weight decay in the radial part only. Additionally, the paper recommends some heuristics for tuning the (radial) learning rate, and prescribes recipes on how to optimize specific deep learning layers (e.g. batch normalisation layers).

Some experiments are reported on Cifar-100 (and modular addition in the Appendix) that compare the new optimization method (AdamO) with Adam, AdamW, as well as ablated versions of AdamO.

**Strengths:**

- The exposition of the method is clear and concise.
- The experimental section presents good results on CIFAR-100 that also backup the claims of the authors regarding reduction of gradient variance.

**Suggestions:**

- A full conference version of this paper should be more standard in terms of references and academic norms. For example, I would discourage the authors from including an emoji in the title of the paper. Furthermore, the paper uses non-standard terms that obfuscate understanding (e.g. "Radial Tug-of-War" in the abstract). Additionally, the first reference (line 032) is completely out of context, and unrelated to the paper. Several sections of the paper appear to have been written with heavy assistance from AI tools. While some parts of the paper appear to be of good quality, others are not, which leaves me on the fence regarding the recommendation.
- I am sceptical about some of the reported results:
     - Figure 1 visualises the decision boundary produced by various optimisation algorithms in a spiral dataset. Why is the training accuracy exactly the same (up to 4th digit) for Adam and AdamW? Why is it larger than the training accuracy of AdamO and Adam0 WD? There is no description of this experiment in the paper as far as I can tell.
    - Figure 4 does not mention the weight decay value for the AdamO ablation study. How did you tune it?

---

### Official Review · Reviewer_4y2t · 2026-02-27

**Fit:** 3
**Significance:** 3
**Confidence:** 1

**Summary:**

This paper proposes an alternative to the weight decay mechanism in AdamW.
The authors redesign a decoupling of weight decay (gradient norm) and feature learning (direction) in the parameter updates to create AdamO.
Experiments show that AdamO learns smoother decision boundaries than AdamW and outperforms it on image (CIFAR-100) and language (Grokking Power) tasks.

**Strengths:**

S1. This paper investigates an interesting phenomenon in neural network training.

S2. The proposed alternative to AdamW, named AdamO, demonstrates strong performance.

S3. I believe this would be of much interest to those working on optimizers.

**Suggestions:**

W1. As an outsider to optimization, many of the technical details are not obvious to me. I would appreciate more exposition and motivation for each expression.

W2. There does not appear to be much theoretical analysis, though this is understandable for a workshop submission.

---

### Meta-Review · Area_Chair_Dr4C · 2026-02-28

**Recommendation:** Accept

**Metareview:**

This paper proposes a modification to to Adam that separates the radial and tangential part of the gradient. While several reviewers noted the clarity and quality of the text could be improved, all reviews noted the experiments demonstrated the effectiveness of the idea. I recommend acceptance.

---

### Decision · Program_Chairs · 2026-03-02

Accept